# Understanding barriers to tuberculosis diagnosis and treatment completion in a low-resource setting: A mixed-methods study in the Kingdom of Lesotho

**Afom T. Andom**[1,2]*, **Hannah N. Gilbert**[2], **Melino Ndayizigiye**[1], **Joia S. Mukherjee**[2,3,4], **Christina Thompson Lively**[2], **Jonase Nthunya**[1], **Tholoana A. Marole**[1], **Makena Ratsiu**[1], **Mary C. Smith Fawzi**[2], **Courtney M. Yuen**[2,3]

1 Partners In Health–Lesotho, Maseru, Lesotho, 2 Department of Global Health and Social Medicine, Harvard Medical School, Boston, MA, United States of America, 3 Division of Global Health Equity, Brigham and Women's Hospital, Boston, MA, United States of America, 4 Partners In Health, Boston, MA, United States of America

* aandom@pih.org

**Data Availability Statement:** All relevant data are within the manuscript and its supporting information files.

## Abstract

### Background

Lesotho is one of the 30 countries with the highest tuberculosis incidence rates in the world, estimated at 650 per 100,000 population. Tuberculosis case detection is extremely low, particularly with the rapid spread of COVID-19, dropping from an estimated 51% in 2020 to 33% in 2021. The aim of this study is to understand the barriers to tuberculosis diagnosis and treatment completion.

### Methods

We used a convergent mixed methods study design. We collected data on the number of clients reporting symptoms upon tuberculosis screening, their sputum test results, the number of clients diagnosed, and the number of clients who started treatment from one district hospital and one health center in Berea district, Lesotho. We conducted in-depth interviews and focus group discussions with 53 health workers and patients. We used a content analysis approach to analyze qualitative data and integrated quantitative and qualitative findings in a joint display.

### Findings

During March-August, 2019, 218 clients at the hospital and 292 clients at the health center reported tuberculosis symptoms. The full diagnostic testing process was completed for 66% of clients at the hospital and 68% at the health center. Among clients who initiated tuberculosis treatment, 68% (61/90) at the hospital and 74% (32/43) at the health center completed treatment. The main barriers to testing and treatment completion were challenges at sample collection, lack of decentralized diagnostic services, and socioeconomic factors such as food insecurity and high patient movement to search for jobs.

**Funding:** This work was conducted with support from the Master of Medical Sciences in Global Health Delivery program of Harvard Medical School Department of Global Health and Social Medicine and financial contributions from Harvard University and the Ronda Stryker and William Johnston MMSc Fellowship in Global Health Delivery. The content is solely the responsibility of the authors and does not necessarily represent the official views of Harvard University and its affiliated academic health care centers. Additional support was provided by Partners In Health – Lesotho. The funders had no role in study design, data collection and analysis, decision to publish, or preparation of the manuscript.

**Competing interests:** The authors have declared that no competing interests exist.

## Conclusions

Tuberculosis diagnosis could be improved through the effective decentralization of laboratory services at the health facility level, and treatment completion could be improved by providing food and other forms of social support to patients.

## Introduction

Tuberculosis remains a major public health threat, with an estimated 10 million new infections and 1.5 million deaths in 2020 [1]. Despite, the availability of effective and free tuberculosis therapeutics for almost eight decades, due to poverty, poor public health systems and the ongoing HIV epidemic, tuberculosis remains one of the top killers among infectious diseases [2,3]. In sub-Saharan African countries where the prevalence of HIV is extremely high, tuberculosis is a particularly important contributor to the burden of disease [4]. It is estimated that 40% of people with tuberculosis are neither diagnosed nor treated globally [1]. Reaching the missed tuberculosis patients, diagnosing them, giving them treatment without delay, and assuring their retention on treatment until they are cured can save millions lives and prevent the spread of tuberculosis [5].

Evaluating the tuberculosis diagnosis and treatment care cascade is a useful approach to improving tuberculosis care delivery; a care cascade analysis helps identify where patients are dropping out of care so that programs can address setting-specific gaps in care seeking, diagnosis and treatment [6]. A study of the tuberculosis care cascades of 30 high tuberculosis burden countries suggested that overall, 35% of people with tuberculosis did not receive a diagnosis while 25% of people who were diagnosed failed to initiate and complete treatment [7]. However, the relative contributions of pre- and post-diagnosis cascade losses differ across countries. In India, it is estimated that 28% of people with tuberculosis never access diagnostic centers, representing the greatest number of people lost from care [8]. However, an evaluation of the South African tuberculosis care cascade showed that only 5% failed to access testing; the care cascade's largest gap was successful treatment completion [9].

In Lesotho, we lack knowledge about the major contributors to gaps in the tuberculosis diagnosis and treatment cascade. Lesotho has the highest tuberculosis incidence rate in the world, with an estimated 650 cases per 100,000 population annually [1]. Despite major efforts by the Government of Lesotho and its partners, only 33% of Lesotho's total estimated tuberculosis cases were diagnosed in 2020, and out of the patients enrolled on tuberculosis treatment only 78% successfully completed their treatment [1]. We previously reported that challenges accessing health facilities and poor implementation of symptom screening within facilities were major contributors to missed diagnoses [10]. In this study, we assess the diagnostic and treatment cascade once individuals with tuberculosis symptoms have been identified in health centers. Understanding and filling the gaps in the tuberculosis diagnosis and care cascade can help to improve tuberculosis care services in Lesotho.

## Materials and methods

### Study design and objective

In this study, we used a convergent mixed methods study design in which qualitative and quantitative data were collected at the same time to complement each other (Fig 1) [11]. Qualitative and quantitative data were analyzed separately, and findings were then integrated for a

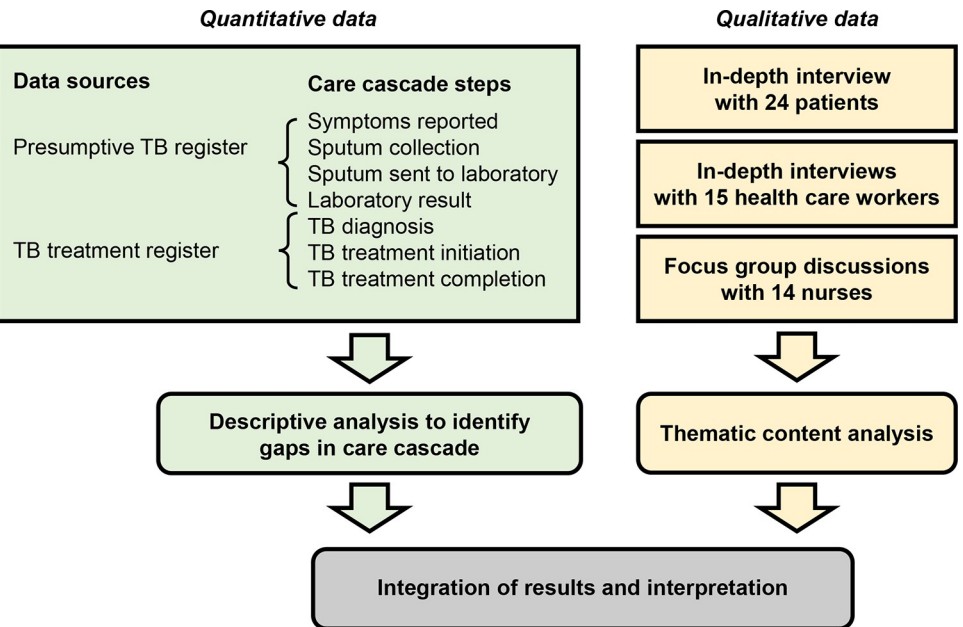

**Fig 1. Study design and data sources.**

deeper understanding of the barriers to completing the tuberculosis care cascade. A previous publication reported reasons for suboptimal tuberculosis screening–the first step of the care cascade [10]. This publication focuses on barriers to completing subsequent steps.

## Study setting

The study was conducted in Berea Hospital, a district-level hospital, and Khubetsoana Health Center, a primary-level facility. Berea Hospital has GeneXpert MTB/RIF, microscopy, and x-ray services for diagnosis of tuberculosis. However, Khubetsoana Health Center does not have any diagnostic facilities, so sputum samples are transported in a scheduled manner twice a week to Maseru Queen Elizabeth II Hospital using a contracted service called Riders for Health. Laboratory results are sent back from the hospital laboratory to the health center, and the health center then contacts patients and requests that they return to the health facility to discuss their results. The turnaround time for laboratory results is, on average, one week at Khubetsoana health center and same-day at Berea Hospital. Both facilities have tuberculosis treatment services.

## Quantitative data collection

We sought to assess completion of key steps of the tuberculosis care cascade for people who attended the two health facilities during March-August 2019. Key steps of the care cascade included completing the symptom screen upon entry to any part of the health facility, collecting sputum if symptoms were present, testing sputum by GeneXpert, recording tuberculosis diagnosis, initiating tuberculosis treatment and completing the prescribed treatment. We previously reported that only 22% of visitors to the hospital and 48% of visitors to the health facility completed tuberculosis symptom screening, which is the first step of the cascade [10].

To assess subsequent steps, we collected data from the presumptive tuberculosis register on the numbers of people who reported symptoms and had sputum collected, with sputum samples sent to the laboratory, with results recorded, and with positive results. We collected data

from treatment registers on the number of people diagnosed with tuberculosis, and people starting and completing tuberculosis treatment. All of these registers were paper-based; we manually counted the aforementioned numbers by month, and entered the data as aggregate numbers into a Microsoft Excel spreadsheet for analysis.

## Quantitative data analysis

We used the care cascade framework [6,12] to identify the steps of the cascade where the most people were being lost. The percentage of people completing each step in the care cascade was calculated by dividing the number of people who completed it by the number who should have completed it. We calculated the percentage of people with sputum collected by dividing the number of sputum samples collected by the number of people who reported any tuberculosis symptom. We calculated the percentage of samples tested by dividing the number of recorded laboratory results by the number of sputum samples sent to the laboratory. We calculated the percentage of tuberculosis patients who started treatment by dividing the number who started treatment by the total number diagnosed with tuberculosis; this denominator included people with a positive laboratory sputum test result as well as those with extrapulmonary tuberculosis and those whose diagnoses were made based on clinical or radiographic criteria. We calculated the percentage of patients who completed treatment by dividing the number who completed treatment by the number who started treatment. All calculations were performed in Microsoft Excel.

## Qualitative data collection

We conducted in-depth interviews with patients and health care workers as well as focus group discussions (FGDs) with nurses in order to understand the processes and experiences that shape loss to follow-up along the TB care cascade in Lesotho. Interview and FGD guides explored screening and diagnostic practices, as well as barriers and facilitators to accessing services and treatment. Separate guides tailored to each participant group were developed by the ATA and HNG, and piloted by the research assistants (NJ, TAM).

We enrolled a total of 53 participants, all of whom were adults (≥18 years old). Participants were purposefully selected to ensure information richness, and to provide multiple perspectives on the factors that shape patient's progression along the cascade [13]. We recruited 24 patients at different stages of the care cascade for in-depth interviews (6 following screening, 6 following diagnosis, 6 undergoing treatment and 6 upon treatment completion). Nurse managers assisted with patient recruitment, and research assistants contacted prospective patient participants by phone. We recruited 15 health care workers representing a range of positions for in-depth interviews. We recruited village health workers, TB screeners, implementing partners, TB coordinator, and lab personnel in person; the lead author recruited the district health manager via email. Finally, we worked with nurse managers to recruit 14 nurses to participate in 2 FGD. While no formal data on refusal was collected, the majority of those contacted elected to participate in the study.

Local research assistants, one male and one female (NJ, TAM), conducted one-time interviews and facilitated the FGDs. Both were university graduates from Lesotho with over 5 years of prior health services research experience, and were trained in qualitative research methods. Research assistants were not previously known to the participants. Research assistants introduced themselves and explained that the research team was interested in understanding participants' experiences with tuberculosis diagnosis and treatment. Interviews and FGD were audio-recorded, and field notes were taken by the research assistants in order to provide context and to support the process of transcription. Interviews and FGDs took place in English

and Sesotho, and were conducted at the study health facilities. FGDs and interviews lasted between 1 and 1.5 hours. Research assistants transcribed and translated the audio recordings. To ensure data quality, transcripts were regularly reviewed by ATA and HNG, and weekly supervisory calls were conducted with the research assistants. Participant feedback on transcripts was not sought for this study.

### Qualitative data analysis

The study team adopted an inductive approach to data analysis, employing a thematic content analytic approach [14] to describe key concepts that identify gaps in the TB care cascade. ATA (a clinician based on Lesotho) open coded a subset of transcripts to identify an initial set of key concepts that were labeled, described, and assembled into a draft codebook which was reviewed by CMY (an epidemiologist) and HNG (a medical anthropologist and qualitative researcher). Discrepancies were resolved through consensus. The codebook was piloted and finalized; the final codebook contained 31 codes. Dedoose version 9.0.54 qualitative data management software was used to support the coding of the complete qualitative dataset. Using an inductive approach, ATA examined the coded data and identified a set of draft themes which were reviewed and revised by CMY and HNG. These more specific themes were grouped into increasingly general concepts, resulting in a set of three thematic categories that explain key gaps in the tuberculosis care cascade. We considered saturation to have occurred when no higher-level concepts emerged. We promoted trustworthiness by collecting data from patients and different types of health care workers, engaging three members of the research team with different backgrounds in the coding process, and providing thick descriptions to contextualize findings within the Lesotho health system.

We assessed how themes identified in qualitative analysis helped to explain gaps in the care cascade uncovered by the quantitative analysis. ATA, HNG, and CMY developed a joint display to integrate qualitative and quantitative findings [15]. While findings were not explicitly shared with patient participants, they were made available to health system participants through presentation at meetings.

### Ethics approval and consent

This study was approved by the National Health Research Ethics Committee of the Kingdom of Lesotho (ID91-2020) and by the Harvard Medical School Institutional Review Board (protocol: IRB20-0109). All people who participated in the interviews and FGDs provided written informed consent.

## Results

### Quantitative care cascade analysis

During March-August 2019, the number of people reporting tuberculosis symptoms was 218 at the hospital and 292 at the health center (Fig 2, S1 Table). Sputum was collected for 94% of people at the hospital and 72% of people at the health center. Over 90% of samples were sent to the laboratory in both facilities. However, at the hospital the number of results recorded was only 76% of the number of samples, while at the health center (which sent its samples to a different hospital laboratory) the number of results was 96% of the number of samples. Thus, the full diagnostic testing process was completed for 66% of people reporting tuberculosis symptoms at the hospital and 68% at the health center. Tuberculosis was detected in 49% of samples from the hospital and 12% of samples from the health center.

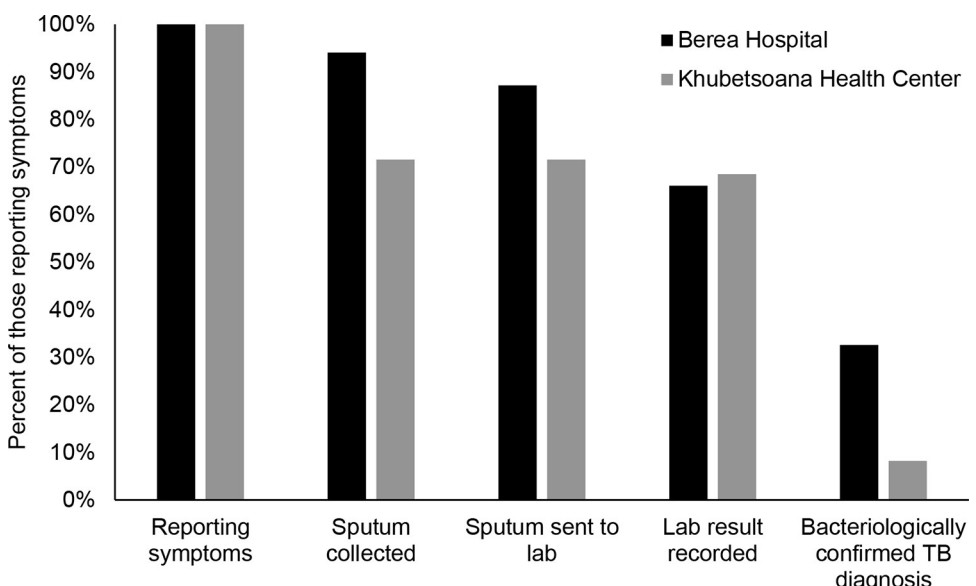

**Fig 2. Tuberculosis (TB) testing cascade for people attending two health facilities during March-August 2019.**

Across both facilities, 134 patients were diagnosed with tuberculosis during the analytic period, with 95 (71%) diagnoses having bacteriologic confirmation. Almost all patients (133/134, 99%) diagnosed with tuberculosis started treatment (Fig 3). The proportion that completed treatment was 68% at the hospital and 74% at the health center.

## Qualitative results

Our qualitative analysis yielded three thematic categories that help explain the low tuberculosis detection and low tuberculosis treatment completion in the Berea district. The thematic categories are: (1) barriers to sputum sample collection from patients; (2) lack of decentralized diagnostic services; (3) and barriers to tuberculosis treatment completion. Fig 4 shows how themes identified in analysis of qualitative data explain quantitative gaps in the care cascade.

**1. Barriers to sputum sample collection from patients.** *a. Timing of sample collection.* Sputum collected at health centers is transported to the district hospital laboratory using a motorcycle transport service called "Riders for Health." Sputum collections occur on specific days of the week, at appointed hours. In most health centers, sputum is collected on just two days per week. If patients arrive at the facility on a date or time that is not in accordance with the Riders' set schedules, patients will be turned away and asked to return at a time that coincides with sputum pick-up. Participants noted that travel to health facilities can be very difficult for patients, and as a result many patients who are turned away may not be able to return. This results in diagnostic delays. Some participants have suggested circumventing this problem by sending Riders for Health directly to villages to collect sputum samples from patients.

*"They [Riders for Health] transport samples on the health centers to the district hospitals and return the results to the health centers. The sample collection is on average twice in a week and the pick the already collected in the morning and they travel to the next health facilities. Patients who come late to the health facilities do not get services in that day and they should wait for the next sample collection day." Nurse, FGD, Berea Hospital*

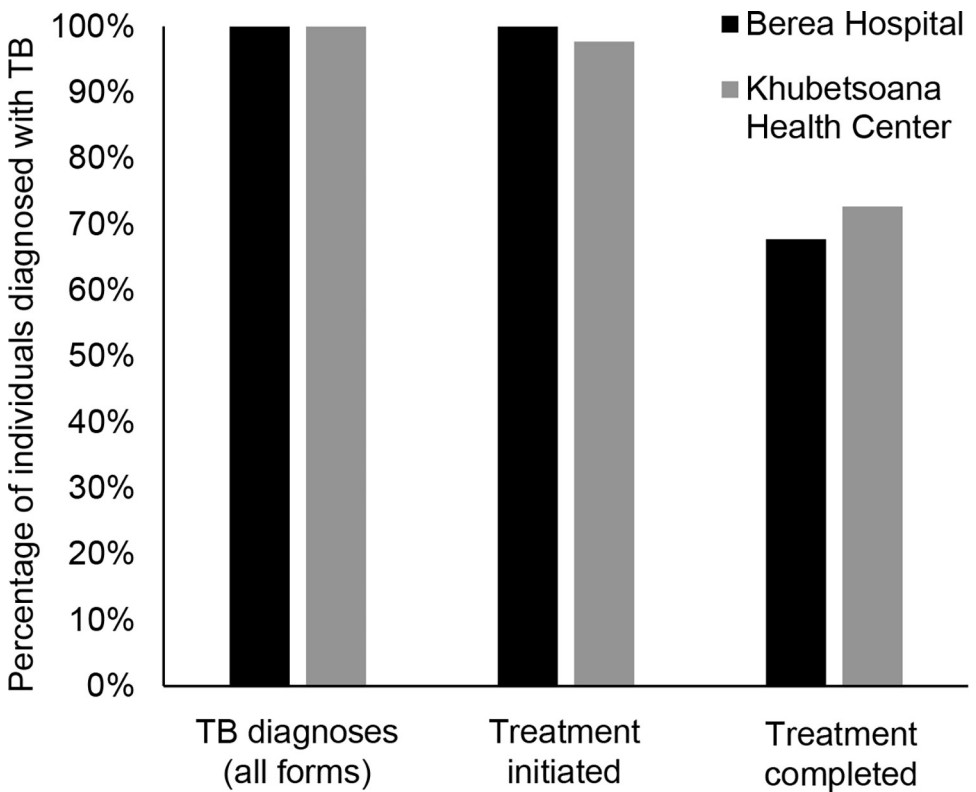

**Fig 3. Tuberculosis (TB) treatment cascade at two health facilities for people diagnosed during March-August 2019.**

*"The first time I left the sputum at the clinic, they gave me empty bottles to take home with me. . . I went back home without the bottles, came back to the health center for another check-up where I was given the bottle again for the sputum—that is the one, I brought to the clinic today." Patient, Khubetsoana Health Center, ID#19*

*"The problem with it is that there are set days for sputum collection so when I come on the day it doesn't come for collection, it means my sputum won't be collected and I will go back home without knowing what my problem is, and there will be a late diagnosis and I am likely to be missed and I might not go back." Health Worker, ID#2*

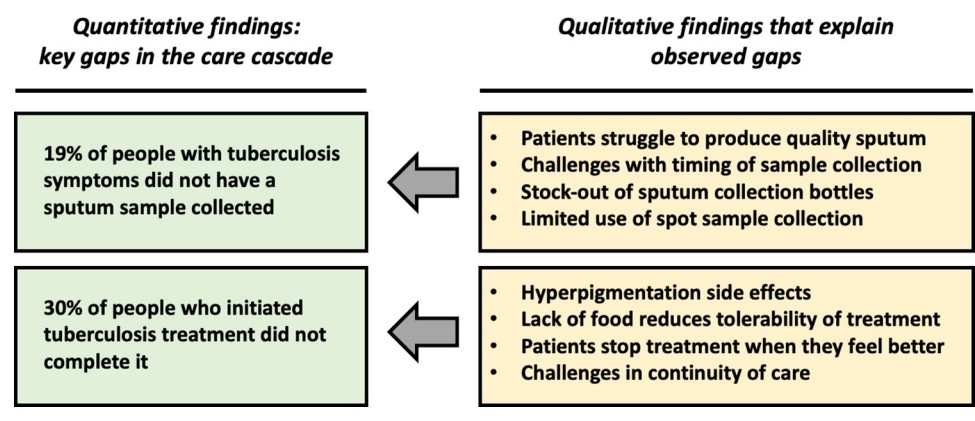

**Fig 4. Joint display showing points of integration between qualitative and quantitative findings.**

*"Some patients may not come to the health facility on the specific days of TB sputum collection due to lack of transport and due to long terrain difficult to travel. It would be good if Riders for Health could be able to travel to the villages and collect the samples from the individual patients." Nurse, FGD, Berea Hospital*

*b. Patients struggle to produce a quality sputum sample.* Health facilities provided insufficient support for patients to provide quality sputum samples. Some participants noted that patients who reported to health facilities were not properly supervised during the provision of sputum. Others explained that while patients were provided with sputum bottles, they were not properly educated on how to produce quality sputum. Unsure of how to proceed, health workers noted that patients would either return with a low-quality sample, or in some cases, would not return at all.

*"Another reason could be because of the sputum quality; a lot of times it is not supervised. A patient is given sputum bottles and instructed to go home and spit and sometimes they bring them back filled with saliva so they will be negative." Health worker, ID#2*

*"I think the facilities should be mentored on how to educate the patient to produce the sputum because some will say, 'I do not have a sputum but I am coughing,' but once you educate the patient surely that patient can leave the facility having left the sputum." Health worker, ID#14*

*"They [Health workers] are not concentrating much to instruct the patient because they have other activities. The patient leaves the facility without understanding how to produce the sputum and how to give sputum in the bottle. The patient decided not to go back to the facility because they fail to understand how to do it." Health Worker, ID#15*

Some patients arrive at the facility at an advanced stage of illness, when they are considerably weakened. Without explicit support or coaching from health workers on proper technique, these patients struggle physically to provide sputum.

*"I think some of them are usually weak when they come to the facility. So, if they have tried to give out the sputum, and there is no sputum no sputum coming out, they just give up and say 'I don't have sputum'- just because they are struggling to give sputum and are also weak to do that." Health worker, ID#1*

*"I don't quite remember what happened because I was sick, but I do remember struggling to spit out the sputum, and I came back to the clinic to tell them that I did not have any sputum." Patient, Khubetsoana Health Center, ID#18*

*c. Stock out of sputum bottles.* Some participants expressed concern that their facilities experience stock outs of sputum bottles. When confronted with a stock out, health workers ask patients to purchase sputum bottles from pharmacies. Many patients at the study facilities were unable to afford the out-of-pocket expenditures associated with purchasing the bottle and traveling to the facility. Participants noted that this problem is compounded by a lack of timely communication related to sputum bottle shortages.

*"If there are no sputum bottles it means the patients will not be able to give sputum because there are no sputum jars. I remember at a certain facility they were even asked to go buy bottles at the pharmacy. Sometimes the patients may be far from the chemist, sometimes they would not have transport to go there, or sometimes they may not have money buy the bottles.*

*And maybe we would learn late that there are no jars when we go to the facility or maybe learn late after being informed by another mentor that there are such challenges in that facility. So I think that this has a bad impact on the TB diagnostic strategies that are not always in place." Health worker, ID#1*

*d. Limited use of spot sample collection.* Spot samples–the first sputum sample collected from a patient–should be collected by a health worker when the patient reports to the facility. Following spot sample collection, patients are then provided with a sputum bottle to capture a second sputum sample at home the following morning, when sputum production is easier. Participants explained that health facility staff were not reliably collecting spot samples from patients when they reported to the health facility. Health workers were sometimes unable to collect these samples because they were too busy with high workloads, or because they lacked the infrastructure to properly store samples. As a result, sputum samples collected at home were effectively the only samples being collected from many patients. Aware of the significant barriers that patients face in returning these samples, some participants recommended that regular, reliable collection of spot samples could improve tuberculosis diagnosis at their facilities.

*"So collecting samples, they [health workers] do not like it. That is why they prefer to give the bottle to take them home and bring the samples. But other times you find that few staff are on duty and no one will go with the patient to instruct on how to give the sample and they prefer to give the bottles and let them go home." Health worker, ID#15*

*"Some of the facilities are still not doing the spot because of the issue that the Riders [for Health] do not come, and the sputum has to stay in the facility for a certain period of time before the Riders come." Health Worker, ID#14*

*"Health facilities are not collecting the spot samples; for any presumptive TB, the spot sample should be collected first and sputum bottle should be given, and the second sample should be collected the next day. But most of the patients are not getting the spot samples. They just get the bottles and go home to bring the sputum the following day. And most of them do not come back because they will be too ill, no transport, the weather may not help them to travel. If we could do spot samples in every health facility, we could detect most of the TB patients while we are waiting for the second sample from home—which may not even come back." Health worker, ID#15*

*"I believe we can do more if we just encourage our patients to leave the spot because that quality spot sputum is the key that we are going to identify this person with. At least if we have it we can trace the patient. If they don't come back but if we have that spot and it is taken to the lab for diagnosis whatever the result it is going to help us track the person when we find out that he/she is indeed a TB patient." Health Worker, ID#14*

**2. Lack of decentralized diagnostic services.** Many participants emphasized the challenges associated with obtaining laboratory and x-ray diagnostic services for patients. Health centers lacked diagnostic equipment, therefore samples had to be transported to district hospitals for processing. If the sputum test result is negative, the patient must travel to the district hospital to receive an x-ray or other laboratory services. Diagnostic delays are further compounded when there are challenges at the hospital, such as malfunctioning equipment.

*"There are no laboratories in the health center. The laboratories are at the hospitals, so that is the difference. So, the difference in getting the results is that those at the hospital get sputum*

*sample results sooner than those at the health center because those at the health center will wait until the riders/bikers get the results to them. This normally takes one day or two days for the results to return or maybe once in a week."* Health Worker, ID#1

*"The clinics don't have machines to diagnose. I think they should have machines and people who can operate them so they can do their own diagnosis. It does not have to be every clinic that has the machines—they can look for a central place, like choose some clinics where these machines can be placed. Bringing every sample here is a problem because this causes a delay in getting the results to the patient because the results have to be taken back to the clinics by the riders and they only get their results then, but this would be made easier by the availability of GeneXpert machines at some other clinics."* Health Worker, ID#3

*"The machines that can check a person's chest for what is wrong in a case where TB isn't the problem. It should also be able to examine the lungs because now we have to go to Queen II [hospital] for that."* Patient, Khubetsoana Health Center, ID#22

*"Also, the patients have to travel far to do x-rays when a patient cannot produce sputum. Patients travel as far as Queen II [hospital] or TY [Berea town] to do x-rays only for them to be told that X-ray machines are not working."* Nurse, FGD, Khubetsoana Health Center

Many participants expressed concern that the lack of diagnostics at the health centers has a significant negative effect on patients. The centralization of these services results in longer diagnostic wait times for patients who use health centers. The lack of diagnostics at the health centers critically affects the provision of quality tuberculosis services, as delayed results hamper providers' ability to develop a patient management plan, and ultimately initiate treatment.

*"I think they should work on giving us our results fast, we shouldn't have to wait for them for a long time. We don't know if we will get them at the time they said we would and we need to know the results so we can get our treatment on time."* Patient, Khubetsoana Health Center, ID#20

*"A patient who cannot produce sputum and we must test using X-ray, sometimes the patient cannot afford to pay for transport and X-ray tests at Queen 2 [hospital]. Sometimes you will find out that the x-ray is not working at Queen 2 therefore end up not being able to start TB treatment for such patient if she/he has TB. This means we will not be able to diagnose such patient. We cannot start giving treatment to such patients because of lack of positive TB diagnosis."* Nurse, FGD, Khubetsoana Health Center

*"The health facilities will be having patients with no results and also the sample will be kept in the hospitals and it will be difficult for them to decide on the patient plan."* Nurse, FGD, Berea Hospital

Many participants suggested that the availability of x-ray and GeneXpert in every health facility would considerably improve tuberculosis detection by reducing the time for diagnosis, and providing additional vital diagnostic inputs. Decentralization of these technologies would eliminate travel barriers for patients, and would allow patients to receive care at trusted facilities that are close to their homes.

*"I think there should be x-rays at every health facility because the sputum results take time, and while waiting for the results the patient is getting sicker. But the x-ray gives the right results immediately and will help us help the patient on time. Sometimes there is something wrong with the sputum the patient has given, and the lab is unable to determine whether they*

*have TB or not and the patient has to come back to the health facility to give another sample and time keeps moving—but with the x-ray it is easy." Health Worker, ID#13*

*"If we could be assisted with an x-ray so that if the patient cannot produce sputum, an x-ray could be used to test TB." Nurse, FGD, Khubetsoana Health Center*

*"I have heard that GeneXpert machine is not that complicated which means they can be installed in every clinic and people capacitated with how to use it so that patients can be tested immediately and given their results." Health Worker, ID#2*

*"I wish the health facilities could have machines so they can be used on us. I was given medication here for my cough and that did not help me and I had to go to a different health facility where the staff and I did not get along. I want to get all my services here." Patient, Khubetsoana Health Center, ID#22*

**3. Barriers to tuberculosis treatment completion.**    *a. Hyperpigmentation side effects of tuberculosis medications*. Some participants explained that tuberculosis medication had the undesirable side effect of changing one's skin color (a potential side effect of the anti-tuberculosis drugs clofazamine and rifampicin). Many participants were concerned about this side effect because they explained that darkening of skin color was associated with tuberculosis and HIV, which are stigmatized.

*"Some still have the fear of testing for TB. Even after testing, they don't deal with it very well. They believe that when one has TB it is because they are HIV-positive so there is that. . . Some still believe that TB drugs have effects on patients' skin, they believe that they darken peoples skin tone. They believe once you start taking them, it becomes obvious that you have them. So, people have the fear of starting TB treatment because it is associated with HIV." Nurse, FGD, Khubetsoana Health Center*

*"There are some patients who stop using them especially patients who are concerned about their skin color. Once their skin color darkens, they decide to stop TB treatment. Not [one] of them will inform us that they have stopped taking drugs, we will only notice this when the sputum stays positive and does not convert to negative that such patient has stopped TB treatment." Nurse, FGD, Khubetsoana Health Center*

*b. Unavailability of food reduces tolerability of treatment*. Patients at the study facilities often grappled with food insecurity, and many hesitated to continue treatment when they could not obtain sufficient food. Some were concerned that taking medication without enough food reduced their ability to physically tolerate their tuberculosis treatment. Others believed that medication had to be taken with food in order the drug to work. For some, tuberculosis medications made them feel hungry–a particularly undesirable side effect for patients facing food insecurity.

*"I always believe that for pills to work well and be effective, they should be taken after having a meal. Since I was weak, I wanted to eat before I took them. Especially after noticing with my father-in-law, I learned a lot from him. He had problems with the treatment, thepills made him to vomit if he took them before eating, so I learned from him that it's only good to take TB pills after eating." Patient, Berea Hospital, ID#7'*

*"Most TB patients come from poor families, so they do not have enough food and do not take treatment when they do not have food in their house." Health Worker, ID# 15*

*"We have problems as people, people who sometimes get affected by this illness, most of the time you find that we get infected and we are suffering from hunger. The TB treatment that we will get needs someone who has food and eats well. At the time that the drugs force you to eat, there should be food."* Patient, Khubetsoana Health Center, ID#16

*"The challenges that I have recognized is that they make you feel very hungry more often. Sometimes that is bad because there are no piece jobs [daily work] at all. You find that you eat a lot in a short space of time, that doesn't sit well with the wife even though she won't say it, but you will recognize her displeasure."* Patient, Berea Hospital, ID#5

Many participants explained that most tuberculosis patients do not have a steady income as most of them do not work, and taking medication without food became toxic to them. The provision of food would make a significant difference on their life. As a result of this, many patients recommended that providing food could play a significant role in helping patients complete treatment.

*"I think an improvement that could be done is that most TB patients do not have means and because of lack of jobs, they should be assisted with food parcels. The food would make a difference. When they are given drugs, you find that there is no food at home, what happens when the patient takes them, and they become toxic. . .I am saying this because I know how these drugs react, for example if I eat two slices of bread and take the drugs after, after a short period I will already feel hungry. So maize meal would be helpful, the drugs need Pap [a local dish]."* Patient, Berea Hospital, ID#5

*"Another challenge is that there are sick people who do not have food and it becomes our responsibility to provide them with food because they can't take their medication without eating. We now have to take money out of our own pockets and buy them food."* Health Worker, ID#12

*"People who test for TB should be given some food packages. This would really help encourage a lot of people to come to the health center to test and it will decrease the number of people with TB. "* Health Worker, ID#10

*c. Challenges in continuity of care.* Treatment adherence was difficult in the context of Lesotho because the region is highly dependent on migratory labor to South Africa's mines. Labor migrants must follow the migratory schedule and often cannot afford to remain in the country to complete their full course of treatment.

*"Because of poverty, most people in Lesotho go to South Africa to search for a job; they cannot stay for six months in the country until they finish their medication."*

*Health Worker, ID# 15*

*"My family had TB, my mother had TB and so did my sister. Unfortunately for my sister it was a problem, she did not complete her treatment because she was not loyal to them. When she was on treatment, she left for South Africa without having completed it. She later got sick again and had to come back home. However, she managed to get cured."* Patient, Khubetsoana Health Center, ID#13

*"Some of our patients they work in South Africa due to lack of job in Lesotho. And they do not finish their TB treatment. They just pick medication for two-three months and they migrate,*

*and they do not come back, I do not know how this problem can be addressed. This is a real gap in TB services in the district.*" Nurse, FGD, Berea Hospital

There are no formal communication systems for contacting highly mobile patients. Facilities do not have accurate contact information for patients, making it difficult for them do patient outreach in cases of loss to follow-up. There is also no standardized system for communication between health facilities, so providers cannot formally convey clinical information when a patient moves and must attend another health facility.

"*There is no transport to track back the defaulted patients in to care. Lack of cell phone to track TB patients who lost follow up or did not appear in their appointment.*" Nurse, FGD, Berea Hospital

"*The weakness will be when the patient might not be followed up due to different reasons like maybe lack of transport or any other things or maybe they have followed up the patient but the patient has moved to another place and is no longer staying at home.*" Health worker, ID#1

"*Most of our patients are very mobile. A patient will be diagnosed in this facility but due to perceptions that it is not TB it is something else, they move to another facility. Hence it is not easy to monitor such patients, resulting in loss to follow up before the treatment is completed. So, we end up having no outcome for that patient.*" Health Worker, ID#14

*d. Patients stop their treatment when they start to feel better.* Multiple participants expressed concern that some patients begin to doubt the validity of their tuberculosis diagnosis when their symptoms start to subside. Some reframe their diagnosis as witchcraft, which would warrant treatment from traditional healer. Participants recounted experiences where patients pause treatment in order to consult traditional healers, only to return to treatment facilities when the disease has progressed to a more complex stage. Other patients elect to stop their treatment when their symptoms improve because they believe that they have been cured. Participants felt that it was critically important to help patients understand that they must remain on treatment even when their symptoms subside.

"*Some of these patients come to the clinic with presumptive TB cases, they get their sputum tested and found to be TB positive and they start TB treatment. After some time on TB treatment, they get advice from people close to them that they do not have TB but rather have been bewitched. That is the time that some of our patients stop their TB treatment.*" Nurse, FGD, Khubetsoana Health Center

"*The drugs make you feel nauseous. One of my neighbors died because of not taking medication. He started treatment and when he got better, he stopped taking them. The problem with him is that when he got better, he went to consult traditional doctors who then advised him to stop taking the medication.*" Patient, Khubetsoana Health Center, ID#13

"*When they start their treatment, they recover quickly. They start to feel better then believe that they have been cured and they stop taking their medication, this is how they are.*" Health Worker, ID# 11

## Discussion

In this study, we found that the entire diagnostic testing cascade was completed for two out of every three people reporting tuberculosis symptoms, and that less than three quarters of people

with tuberculosis successfully completed treatment. For people identified with tuberculosis symptoms, the major drop-off points in the diagnostic cascade were the collection of sputum samples in the health centers and the testing of sputum samples at the hospital. Challenges during tuberculosis sample collection and lack of decentralized diagnostic services contributed to gaps in the diagnostic cascade, while clinical, social, and health system factors all posed barriers to tuberculosis treatment completion. The lack of laboratory services at the health centers and lack of specialized laboratory services to diagnose extrapulmonary tuberculosis likely contributed to missing around half of the expected tuberculosis cases in Khubetsoana Health Center based on the size of the catchment population and the national tuberculosis incidence estimate [16].

The challenges that participants described around sputum collection help to explain why only 72% of people with tuberculosis symptoms at the health center had sputum collected. Similar to our study, studies from other sub-Saharan African settings have identified lack of coaching patients through sputum collection [17,18], lack of sputum collection supplies [19], and limited capacity for transporting sputum samples from peripheral centers to centralized laboratories [18,19] as barriers to sputum testing. Ideally, high-quality sputum samples should be collected from all individuals with possible tuberculosis symptoms and transported to a laboratory in a timely manner. Tools such as instructional videos for sputum collection could potentially be used to improve sputum sample quality without overburdening health care workers [20]. Strengthening of specimen transport networks (e.g. to allow daily transport) could improve sputum testing, but decentralized tuberculosis diagnostic capacity would be preferable.

Indeed, our qualitative findings suggest that the lack of laboratory services at health centers contributes to diagnostic delays and missed diagnosis in Lesotho, similar to what has been found in other settings [21,22]. Patient expenditures may also increase when patients are forced to visit multiple times to various health facilities to get a tuberculosis diagnosis [23] or when they must travel to a regional facility to access tuberculosis diagnostic services [21]. Increased diagnostic capacity and decentralization of testing to the peripheral health centers is essential for improving tuberculosis detection. Our findings revealed that it is not only laboratory services but also radiology services that must be decentralized for the early diagnosis and detection of tuberculosis [24]. Allocating enough resources to build the tuberculosis diagnostic system and robust laboratory supply chain management is critical for controlling tuberculosis epidemics.

Of the various factors that affect tuberculosis treatment completion, one of the most important and addressable was lack of food. Other studies have similarly found that patients are reluctant to take medication on an empty stomach, feeling that doing so will cause pain or worsen side effects, and that lack of food thus poses a barrier to adherence [25,26]. Tuberculosis primarily attacks the most economically vulnerable segment of societies. The adverse effect of tuberculosis's financial consequences damages the well-being of patients and their families [27], which can exacerbate food scarcity. Provision of financial support and other forms of social protection for patients with limited resources is a proven strategy to improve treatment outcomes [28,29].

This study has some limitations. Firstly, the study was conducted in one district hospital and a primary health center in one district. As a result, our findings may be less generalizable. However, we tried to overcome the limitation of our quantitative component by conducting interviews and focus group discussions with 53 participants, including patients and health care workers from different levels of health sectors. Secondly, the data source for our quantitative component was paper-based records, which limited the amount of data we could feasibly collect and prevented us from assessing the impact of patient characteristics such as HIV on care cascade completion.

## Conclusions

Although in our setting, the greatest loss from the care cascade was at the initial step of screening people for tuberculosis symptoms [10], lack of decentralization of laboratory services at the health center was also a major contributor to low tuberculosis detection. Moreover, we found that patients may fail to complete their treatment successfully due to poverty and socioeconomic factors. Timely tuberculosis diagnosis and immediate commencement of treatment are of critical importance to controlling the spread of tuberculosis. Building a robust health system in which all levels—including community, facility, district, and national levels—are well equipped with reliable tuberculosis diagnosis testing can help to halt the spread of tuberculosis epidemics. As tuberculosis primarily affects the marginalized and impoverished segment of the population, protecting tuberculosis patients and their families from financial hardship through social support such as food and transport assistance is critically helpful for controlling and eliminating tuberculosis in Lesotho and beyond.

## Supporting information

**S1 Checklist. COREQ (COnsolidated criteria for REporting Qualitative research) checklist.**
(PDF)

**S2 Checklist.**
(DOCX)

**S1 Table. Numbers of individuals completing each step of the care cascade at two health facilities.**
(DOCX)

**S1 File.**
(DOCX)

## Author Contributions

**Conceptualization:** Afom T. Andom, Hannah N. Gilbert, Joia S. Mukherjee, Mary C. Smith Fawzi, Courtney M. Yuen.

**Formal analysis:** Afom T. Andom, Hannah N. Gilbert, Courtney M. Yuen.

**Investigation:** Jonase Nthunya, Tholoana A. Marole, Makena Ratsiu.

**Project administration:** Afom T. Andom, Melino Ndayizigiye, Christina Thompson Lively, Makena Ratsiu.

**Resources:** Melino Ndayizigiye, Joia S. Mukherjee, Christina Thompson Lively.

**Supervision:** Afom T. Andom, Hannah N. Gilbert, Melino Ndayizigiye, Christina Thompson Lively, Courtney M. Yuen.

**Writing – original draft:** Afom T. Andom, Hannah N. Gilbert, Courtney M. Yuen.

**Writing – review & editing:** Afom T. Andom, Hannah N. Gilbert, Melino Ndayizigiye, Joia S. Mukherjee, Christina Thompson Lively, Jonase Nthunya, Tholoana A. Marole, Makena Ratsiu, Mary C. Smith Fawzi, Courtney M. Yuen.

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
