## [Decision Letter · Decision Letter 0]

27 Feb 2023

PONE-D-22-29321Understanding Barriers to Tuberculosis Diagnosis and Treatment Completion in a Low-Resource Setting: A Mixed-Methods Study in the Kingdom of LesothoPLOS ONE

Dear Dr. Yuen,

Thank you for submitting your manuscript to PLOS ONE. After careful consideration, we feel that it has merit but does not fully meet PLOS ONE’s publication criteria as it currently stands. Therefore, we invite you to submit a revised version of the manuscript that addresses the points raised during the review process.

We look forward to receiving your revised manuscript.

Kind regards,

Nelsensius Klau Fauk, S.Fil., M., MHID, MSc, PhD

Academic Editor

PLOS ONE

Journal Requirements:

2. You indicated that you had ethical approval for your study. Please clarify whether minors (participants under the age of 18 years) were included in this study. If yes, in your Methods section, please ensure you have also stated whether you obtained consent from parents or guardians of the minors included in the study or whether the research ethics committee or IRB specifically waived the need for their consent.

3. Please include a complete copy of PLOS’ questionnaire on inclusivity in global research in your revised manuscript. Our policy for research in this area aims to improve transparency in the reporting of research performed outside of researchers’ own country or community. The policy applies to researchers who have travelled to a different country to conduct research, research with Indigenous populations or their lands, and research on cultural artefacts. The questionnaire can also be requested at the journal’s discretion for any other submissions, even if these conditions are not met.  Please find more information on the policy and a link to download a blank copy of the questionnaire here: https://journals.plos.org/plosone/s/best-practices-in-research-reporting. Please upload a completed version of your questionnaire as Supporting Information when you resubmit your manuscript.

This work was conducted with support from the Master of Medical Sciences in Global Health Delivery program of Harvard Medical School Department of Global Health and Social Medicine and financial contributions from Harvard University and the Ronda Stryker and William Johnston MMSc Fellowship in Global Health Delivery. The content is solely the responsibility of the authors and does not necessarily represent the official views of Harvard University and its affiliated academic health care centers.  Additional support was provided by Partners In Health – Lesotho.

However, funding information should not appear in the Acknowledgments section or other areas of your manuscript. We will only publish funding information present in the Funding Statement section of the online submission form. 

This work was conducted with support from the Master of Medical Sciences in Global Health Delivery program of Harvard Medical School Department of Global Health and Social Medicine and financial contributions from Harvard University and the Ronda Stryker and William Johnston MMSc Fellowship in Global Health Delivery. The content is solely the responsibility of the authors and does not necessarily represent the official views of Harvard University and its affiliated academic health care centers.   Additional support was provided by Partners In Health – Lesotho. The funders had no role in study design, data collection and analysis, decision to publish, or preparation of the manuscript.

Additional Editor Comments:

Please address the reviewers' comments carefully and resubmit.

Reviewers' comments:

Reviewer's Responses to Questions

**Comments to the Author**

1. Is the manuscript technically sound, and do the data support the conclusions?

Reviewer #1: Partly

Reviewer #2: Yes

2. Has the statistical analysis been performed appropriately and rigorously? 

Reviewer #1: Yes

Reviewer #2: Yes

3. Have the authors made all data underlying the findings in their manuscript fully available?

Reviewer #1: Yes

Reviewer #2: Yes

4. Is the manuscript presented in an intelligible fashion and written in standard English?

Reviewer #1: Yes

Reviewer #2: Yes

5. Review Comments to the Author

Reviewer #1: PONE-D-22-29321

Understanding Barriers to Tuberculosis Diagnosis and Treatment Completion in a Low Resource Setting: A Mixed-Methods Study in the Kingdom of Lesotho

Comments

Thank you for having me review this exciting study.

- Introduction: The introduction is too short and cites a reference from WHO (reference number 1. The introduction needs to show evidence or what has been known about the problem. For example, are there any other studies that explored this issue before? Although it came from another study setting or country, previously published studies can be used as a “bible” in conducting new research and later show the novelty of the current study compared to the others.

- Introduction: The layout needs to be developed as every paragraph mentions tuberculosis epidemiology.

- Introduction: Line 65 “As a first step….” Elaborate more; is it a part of another study? Not sure about “the first step,” as the authors also mentioned other publications (lines 86-88).

- Methods: What was the reporting guideline used? Please refers to: https://www.equator-network.org/ or Good Reporting of A Mixed Methods Study (GRAMMS) Checklist

- Methods: The sequence can be reorganized. Probably start with the quantitative part: including the purpose, participants’ recruitment, and tools, …. Up to the data analysis steps. After that, then the qualitative approach can be explained in a similar sequence.

- Methods: For the qualitative part, some information is missing: (1) who did the interviews (not limited to the gender only, but also their experience, expertise, experience, and training in qualitative?), (2) “The interviews and FGDs took place in English and Sesoho..” What does it mean? (3), The sequence of the explanation needs to be revised: the author started directly with “2 FGDs with nurses…” and later talked about the recruitment process and interview guidelines… (4) What did the authors do to ensure the trustworthiness of the study? (5) Who did the data analysis? (not only the initials, but also the credentials of the researchers who did the data analysis), and (5) How did the team decide on the data saturation?

- Methods: The power of mixed methods is the integration of qualitative and quantitative approaches, whatever the mixed methods approach they used (in this paper, it’s a convergent methods). But I did not see why they needed to use a mixed form (it would be helpful to be explained in the introduction or after they mentioned the “convergent” part)? In the method’s part, I did not see how they compared or integrated quantitative and qualitative data analysis to see if the data confirmed or disconfirmed each other.

See: Creswell, J.W., & Creswell, J.D. (2018). Mixed methods procedures. In, Research design: Qualitative, quantitative, and mixed methods approaches (5th ed., pp. 213-246). Los Angeles, CA: SAGE Publications, Inc.

- Results: The quantitative and qualitative results have been presented. Again, as this is a mixed methods, I did not see the data integration of both sides. The two results should be merged in the interpretation stage.

Reviewer #2: PONE-D-22-29321

Understanding Barriers to Tuberculosis Diagnosis and Treatment Completion in a Low Resource Setting: A Mixed-Methods Study in the Kingdom of Lesotho

This manuscript reports very important and useful findings for the improvement of tuberculosis diagnosis and treatment. However, I have several comments that need to be addressed to improve the manuscript.

Introduction

The main missing aspect in the introduction section is the lack of synthesis of the literature/existing findings on both barriers to tuberculosis diagnosis and barriers to tuberculosis treatment completion. Based on your synthesis of the existing findings from previous studies, the authors can then strongly state the novelty of their findings.

The authors stated “only 33% of Lesotho’s total estimated tuberculosis cases were diagnosed in 2020, and out of the patients enrolled on tuberculosis treatment only 78% successfully completed their treatment” but this doesn’t say much about the in knowledge. This the context in the study setting, which is a strong reason why they wanted to explore the barriers.

Methods

For the qualitative section: I suggest the authors check the COREQ checklist to guide the transparent reporting of the method section.

Who did the interviews? What qualifications do the interviewer(s) have? What language was used and why? Do they use an interview guide? What about data saturation? Were the interviews/FGDs audio-recorded? If not, how could you remember all the conversations during the interviews/FGDs? I know for sure it is impossible to remember everything verbatim.

Who transcribe the audio recordings? Who did the translation into English, if the language used was not English?

Discussion

Please ensure that you discuss your findings in light of previous findings on the same topic as yours and the related theories.

6. PLOS authors have the option to publish the peer review history of their article (what does this mean?). If published, this will include your full peer review and any attached files.

Reviewer #1: No

Reviewer #2: **Yes: **Nelsensius Klau Fauk

---

## [Author Response · Author response to Decision Letter 0]

23 Mar 2023

We have uploaded point-by-point responses to reviewer and editor comments as a separate document. They are replicated here without formatting (but probably easier to read in the formatted version).

Editorial comments

Comment 1. Please ensure that your manuscript meets PLOS ONE's style requirements, including those for file naming. The PLOS ONE style templates can be found at 

Response: We have formatted the document according to the guidance provided

Comment 2. You indicated that you had ethical approval for your study. Please clarify whether minors (participants under the age of 18 years) were included in this study. If yes, in your Methods section, please ensure you have also stated whether you obtained consent from parents or guardians of the minors included in the study or whether the research ethics committee or IRB specifically waived the need for their consent.

Response: All participants were adults at least 18 years old. We have added this statement to the methods.

Comment 3. Please include a complete copy of PLOS’ questionnaire on inclusivity in global research in your revised manuscript. Our policy for research in this area aims to improve transparency in the reporting of research performed outside of researchers’ own country or community. The policy applies to researchers who have travelled to a different country to conduct research, research with Indigenous populations or their lands, and research on cultural artefacts. The questionnaire can also be requested at the journal’s discretion for any other submissions, even if these conditions are not met. Please find more information on the policy and a link to download a blank copy of the questionnaire here: https://journals.plos.org/plosone/s/best-practices-in-research-reporting. Please upload a completed version of your questionnaire as Supporting Information when you resubmit your manuscript.

Response: We have uploaded the questionnaire.

Comment 4. Thank you for stating the following in the Acknowledgments Section of your manuscript: 

This work was conducted with support from the Master of Medical Sciences in Global Health Delivery program of Harvard Medical School Department of Global Health and Social Medicine and financial contributions from Harvard University and the Ronda Stryker and William Johnston MMSc Fellowship in Global Health Delivery. The content is solely the responsibility of the authors and does not necessarily represent the official views of Harvard University and its affiliated academic health care centers. Additional support was provided by Partners In Health – Lesotho.

However, funding information should not appear in the Acknowledgments section or other areas of your manuscript. We will only publish funding information present in the Funding Statement section of the online submission form. 

This work was conducted with support from the Master of Medical Sciences in Global Health Delivery program of Harvard Medical School Department of Global Health and Social Medicine and financial contributions from Harvard University and the Ronda Stryker and William Johnston MMSc Fellowship in Global Health Delivery. The content is solely the responsibility of the authors and does not necessarily represent the official views of Harvard University and its affiliated academic health care centers. Additional support was provided by Partners In Health – Lesotho. The funders had no role in study design, data collection and analysis, decision to publish, or preparation of the manuscript.

Response: Thank you, we have removed the acknowledgements section from the manuscript file. The funding statement is correct as written, and we request no changes.

Reviewer 1 comments

Comment 1: Introduction: The introduction is too short and cites a reference from WHO (reference number 1. The introduction needs to show evidence or what has been known about the problem. For example, are there any other studies that explored this issue before? Although it came from another study setting or country, previously published studies can be used as a “bible” in conducting new research and later show the novelty of the current study compared to the others.

Response: We have revised the introduction to better contextualize the study. The second paragraph now describes care cascade assessments from different countries and why they are important for revealing setting-specific gaps in care. The third paragraph now explains what is and is not known about the care cascade in Lesotho, and hence what this study adds to the literature.

Comment 2: Introduction: The layout needs to be developed as every paragraph mentions tuberculosis epidemiology.

Response: We have reduced the focus on tuberculosis epidemiology in the introduction, removing several sentences about global and Lesotho-specific epidemiologic measures, and adding new material to the second and third paragraphs as described in the response to Comment 1.

Comment 3: Introduction: Line 65 “As a first step….” Elaborate more; is it a part of another study? Not sure about “the first step,” as the authors also mentioned other publications (lines 86-88).

Response: We agree this was confusing and have removed the phrase.

Comment 4: Methods: What was the reporting guideline used? Please refers to: https://www.equator-network.org/ or Good Reporting of A Mixed Methods Study (GRAMMS) Checklist

Response: We have provided the COREQ checklist (as suggested by reviewer 2) with this revision.

Comment 5: Methods: The sequence can be reorganized. Probably start with the quantitative part: including the purpose, participants’ recruitment, and tools, …. Up to the data analysis steps. After that, then the qualitative approach can be explained in a similar sequence.

Response: We have reordered the methods to state the overall study design and objective first. Then, within each section (quantitative/qualitative), we start with a statement of purpose, describe the participants, the data collection, and the analysis. We have also added additional paragraph breaks to more clearly break out each of these topics.

Comment 6: Methods: For the qualitative part, some information is missing: (1) who did the interviews (not limited to the gender only, but also their experience, expertise, experience, and training in qualitative?), (2) “The interviews and FGDs took place in English and Sesoho..” What does it mean? (3), The sequence of the explanation needs to be revised: the author started directly with “2 FGDs with nurses…” and later talked about the recruitment process and interview guidelines… (4) What did the authors do to ensure the trustworthiness of the study? (5) Who did the data analysis? (not only the initials, but also the credentials of the researchers who did the data analysis), and (5) How did the team decide on the data saturation?

Response: 

(1) We have added the qualifications of the interview facilitators: “Both were university graduates from Lesotho with over 5 years of prior health services research experience, and were trained in qualitative research methods.”

(2) We are reporting the languages used to conduct the interviews. We have tried to clarify this by revising to “The interviews and FGDs were conducted in English and Sesotho”

(3) We have reordered the qualitative methods to describe the purpose of the interviews/FGD and the content of the interview guides, then participant recruitment, then interview/FGD methods.

(4) Trustworthiness was promoted by collecting data from patients and different types of health care workers, engaging three members of the research team with different backgrounds in the coding process, and providing thick descriptions to contextualize findings within the Lesotho health system

(5) We have moved the description of the investigators’ credentials to the section on data analysis.

(6) We have added the second sentence in the following passage, “These more specific themes were grouped into increasingly general concepts, resulting in a set of three thematic categories that explain key gaps in the tuberculosis care cascade. We considered saturation to have occurred when no higher-level concepts emerged”

Comment 7: Methods: The power of mixed methods is the integration of qualitative and quantitative approaches, whatever the mixed methods approach they used (in this paper, it’s a convergent methods). But I did not see why they needed to use a mixed form (it would be helpful to be explained in the introduction or after they mentioned the “convergent” part)? In the method’s part, I did not see how they compared or integrated quantitative and qualitative data analysis to see if the data confirmed or disconfirmed each other.

See: Creswell, J.W., & Creswell, J.D. (2018). Mixed methods procedures. In, Research design: Qualitative, quantitative, and mixed methods approaches (5th ed., pp. 213-246). Los Angeles, CA: SAGE Publications, Inc.

Response: We have added the following statement to the end of the methods: “We assessed how themes identified in qualitative analysis helped to explain gaps in the care cascade uncovered by the quantitative analysis. We used a joint display to integrate qualitative and quantitative findings [Guetterman et al, 2015].”

Comment 8: Results: The quantitative and qualitative results have been presented. Again, as this is a mixed methods, I did not see the data integration of both sides. The two results should be merged in the interpretation stage.

Response: We have added Figure 4, a joint display to show the integration of the quantitative and qualitative findings.

Reviewer 2 comments

Comment 1: Introduction - The main missing aspect in the introduction section is the lack of synthesis of the literature/existing findings on both barriers to tuberculosis diagnosis and barriers to tuberculosis treatment completion. Based on your synthesis of the existing findings from previous studies, the authors can then strongly state the novelty of their findings.

The authors stated “only 33% of Lesotho’s total estimated tuberculosis cases were diagnosed in 2020, and out of the patients enrolled on tuberculosis treatment only 78% successfully completed their treatment” but this doesn’t say much about the in knowledge. This the context in the study setting, which is a strong reason why they wanted to explore the barriers.

Response: We have revised the introduction to better contextualize the study. The second paragraph now describes care cascade assessments from different countries and why they are important for revealing setting-specific gaps in care. The third paragraph now explains what is and is not known about the care cascade in Lesotho, and hence what this study adds to the literature.

Comment 2: Methods - For the qualitative section: I suggest the authors check the COREQ checklist to guide the transparent reporting of the method section. Who did the interviews? What qualifications do the interviewer(s) have? What language was used and why? Do they use an interview guide? What about data saturation? Were the interviews/FGDs audio-recorded? If not, how could you remember all the conversations during the interviews/FGDs? I know for sure it is impossible to remember everything verbatim. Who transcribe the audio recordings? Who did the translation into English, if the language used was not English?

Response: We have completed the COREQ checklist and submitted it with this revision. We have added the interviewers’ qualifications (“Both were university graduates from Lesotho with over 5 years of prior health services research experience, and were trained in qualitative research methods”). The languages were specified (“The interviews and FGDs were conducted in English and Sesotho”). We have added that the interviews and FGD were audio-recorded, and that the research assistants did the transcription and translation.

Comment 3: Discussion Please ensure that you discuss your findings in light of previous findings on the same topic as yours and the related theories.

Response: We have revised the discussion in the following ways:

- We have refocused the discussion on three specific findings: challenges in sputum collection (new), the need for decentralized laboratory capacity (present in the original discussion), and the role of food scarcity in reducing adherence (new)

- For each, we highlight prior studies from different countries where similar issues were described

- We suggest what types of program improvements are necessary to overcome the barriers that our study uncovered and cite examples of prior studies that have demonstrated successful strategies.

The relevant paragraphs are reproduced below:

The challenges that participants described around sputum collection help to explain why only 72% of people with tuberculosis symptoms at the health center had sputum collected. Similar to our study, studies from other sub-Saharan African settings have identified lack of coaching patients through sputum collection [17, 18], lack of sputum collection supplies [19], and limited capacity for transporting sputum samples from peripheral centers to centralized laboratories [18, 19] as barriers to sputum testing. Ideally, high-quality sputum samples should be collected from all individuals with possible tuberculosis symptoms and transported to a laboratory in a timely manner. Tools such as instructional videos for sputum collection could potentially be used to improve sputum sample quality without overburdening health care workers [20]. Strengthening of specimen transport networks (e.g. to allow daily transport) could improve sputum testing, but decentralized tuberculosis diagnostic capacity would be preferable.

Indeed, our qualitative findings suggest that the lack of laboratory services at health centers contributes to diagnostic delays and missed diagnosis in Lesotho, similar to what has been found in other settings [21, 22]. Patient expenditures may also increase when patients are forced to visit multiple times to various health facilities to get a tuberculosis diagnosis [23] or when they must travel to a regional facility to access tuberculosis diagnostic services [21]. Increased diagnostic capacity and decentralization of testing to the peripheral health centers is essential for improving tuberculosis detection. Our findings revealed that it is not only laboratory services but also radiology services that must be decentralized for the early diagnosis and detection of tuberculosis [24]. Allocating enough resources to build the tuberculosis diagnostic system and robust laboratory supply chain management is critical for controlling tuberculosis epidemics. 

Of the various factors that affect tuberculosis treatment completion, one of the most important and addressable was lack of food. Other studies have similarly found that patients are reluctant to take medication on an empty stomach, feeling that doing so will cause pain or worsen side effects, and that lack of food thus poses a barrier to adherence [25, 26]. Tuberculosis primarily attacks the most economically vulnerable segment of societies. The adverse effect of tuberculosis's financial consequences damages the well-being of patients and their families [27], which can exacerbate food scarcity. Provision of financial support and other forms of social protection for patients with limited resources is a proven strategy to improve treatment outcomes [28, 29].

---

## [Decision Letter · Decision Letter 1]

27 Apr 2023

PONE-D-22-29321R1Understanding barriers to tuberculosis diagnosis and treatment completion in a low-resource setting: a mixed-methods study in the Kingdom of LesothoPLOS ONE

Dear Dr. Yuen,

Thank you for submitting your manuscript to PLOS ONE. After careful consideration, we feel that it has merit but does not fully meet PLOS ONE’s publication criteria as it currently stands. Therefore, we invite you to submit a revised version of the manuscript that addresses the points raised during the review process.

We look forward to receiving your revised manuscript.

Kind regards,

Nelsensius Klau Fauk, S.Fil., M., MHID, MSc, PhD

Academic Editor

PLOS ONE

Journal Requirements:

Reviewers' comments:

Reviewer's Responses to Questions

**Comments to the Author**

1. If the authors have adequately addressed your comments raised in a previous round of review and you feel that this manuscript is now acceptable for publication, you may indicate that here to bypass the “Comments to the Author” section, enter your conflict of interest statement in the “Confidential to Editor” section, and submit your "Accept" recommendation.

Reviewer #1: All comments have been addressed

Reviewer #2: All comments have been addressed

2. Is the manuscript technically sound, and do the data support the conclusions?

Reviewer #1: Yes

Reviewer #2: Yes

3. Has the statistical analysis been performed appropriately and rigorously? 

Reviewer #1: Yes

Reviewer #2: Yes

4. Have the authors made all data underlying the findings in their manuscript fully available?

Reviewer #1: Yes

Reviewer #2: Yes

5. Is the manuscript presented in an intelligible fashion and written in standard English?

Reviewer #1: Yes

Reviewer #2: Yes

6. Review Comments to the Author

Reviewer #1: Re: Understanding barriers to tuberculosis diagnosis and treatment completion in a lowresource setting: a mixed-methods study in the Kingdom of Lesotho

PONE-D-22-29321R1

Thank you for allowing us to review the authors' responses. The authors generally responded to all of my comments. Nonetheless, I have a few concerns:

Abstract

- Background and methods should be re-writing to avoid the redundancy of the information (“…to understand the barriers….”). Better to use the space to describe the analysis methods for both quantitative and qualitative designs.

- Findings: sample size should be mentioned before presenting the percentage. There is no information regarding the number of participants for the IDIs and FGDs.

- I am questioning the conclusion: "Providing social support such as food and transport support could prevent patients and families from experiencing financial hardship and help patients complete their treatment successfully”. The findings stated “food insecurity and high patient movement to search for jobs”. And the conclusion “…providing food and transport support… to complete their treatment”. So the transport barriers are related to “job seeking” and the conclusion is related to “the completion of the treatment”. Although it can be related indirectly, but not sure the authors can relate this based on a qualitative study.

Reporting guidelines:

Using a particular reporting guideline will improve the quality of the manuscripts. I agree with using COREQ as the reporting guidelines. However, COREQ is “Consolidated criteria for reporting qualitative research”; therefore, it is intended for a qualitative report. How about the quantitative findings? That’s why I suggested using GRAMMS, for example, or other reporting guidelines suitable for a mixed method.

Reviewer #2: (No Response)

7. PLOS authors have the option to publish the peer review history of their article (what does this mean?). If published, this will include your full peer review and any attached files.

Reviewer #1: No

Reviewer #2: No

---

## [Author Response · Author response to Decision Letter 1]

27 Apr 2023

We have made the following changes in response to the reviewer’s comments.

Comments about the abstract:

Comment: Background and methods should be re-writing to avoid the redundancy of the information (“…to understand the barriers….”). Better to use the space to describe the analysis methods for both quantitative and qualitative designs.

Response: We have deleted the redundant phrase from the methods section and added details of the analysis: “We used a content analysis approach to analyze qualitative data and integrated quantitative and qualitative findings in a joint display.”

Comment: Findings: sample size should be mentioned before presenting the percentage. There is no information regarding the number of participants for the IDIs and FGDs.

Response: We have added all denominators for the quantitative results: “ During March-August, 2019, 218 clients at the hospital and 292 clients at the health center reported tuberculosis symptoms. The full diagnostic testing process was completed for 66% of clients at the hospital and 68% at the health center. Among clients who initiated tuberculosis treatment 68% (61/90) at the hospital and 74% (32/43) at the health center completed treatment.” We have added the number of participants in interviews and focus groups to the methods: “We conducted in-depth interviews and focus group discussions with 53 health workers and patients.”

Comment: I am questioning the conclusion: "Providing social support such as food and transport support could prevent patients and families from experiencing financial hardship and help patients complete their treatment successfully”. The findings stated “food insecurity and high patient movement to search for jobs”. And the conclusion “…providing food and transport support… to complete their treatment”. So the transport barriers are related to “job seeking” and the conclusion is related to “the completion of the treatment”. Although it can be related indirectly, but not sure the authors can relate this based on a qualitative study.

Response: We have revised the conclusion to eliminate the reference to transport: “Tuberculosis diagnosis could be improved through the effective decentralization of laboratory services at the health facility level, and treatment completion could be improved by providing food and other forms of social support to patients.”

Comment about reporting guidelines:

Comment: Using a particular reporting guideline will improve the quality of the manuscripts. I agree with using COREQ as the reporting guidelines. However, COREQ is “Consolidated criteria for reporting qualitative research”; therefore, it is intended for a qualitative report. How about the quantitative findings? That’s why I suggested using GRAMMS, for example, or other reporting guidelines suitable for a mixed method.

Response: We have uploaded the GRAMMS table in addition to the COREQ.

---

## [Editor Report · Decision Letter 2]

2 May 2023

Understanding barriers to tuberculosis diagnosis and treatment completion in a low-resource setting: a mixed-methods study in the Kingdom of Lesotho

PONE-D-22-29321R2

Dear Dr. Yuen,

We’re pleased to inform you that your manuscript has been judged scientifically suitable for publication and will be formally accepted for publication once it meets all outstanding technical requirements.

Kind regards,

Nelsensius Klau Fauk, S.Fil., M., MHID, MSc, PhD

Academic Editor

PLOS ONE
---

## [Editor Report · Acceptance letter]

4 May 2023

PONE-D-22-29321R2 

Understanding barriers to tuberculosis diagnosis and treatment completion in a low-resource setting: a mixed-methods study in the Kingdom of Lesotho 

Dear Dr. Yuen:

I'm pleased to inform you that your manuscript has been deemed suitable for publication in PLOS ONE. Congratulations! Your manuscript is now with our production department. 

Kind regards, 

on behalf of

Dr. Nelsensius Klau Fauk 

Academic Editor

PLOS ONE